# Eastern Grey Kangaroo (*Macropus giganteus*) Vigilance Behaviour Varies between Human-Modified and Natural Environments

**DOI:** 10.3390/ani9080494

**Published:** 2019-07-27

**Authors:** Georgina Hume, Elizabeth Brunton, Scott Burnett

**Affiliations:** 1University of the Sunshine Coast, 91 Sippy Downs Drive, Sippy Downs, QLD 4556, Australia; 2Global-Change Ecology Research Group, School of Science and Engineering, University of the Sunshine Coast, Sippy Downs, QLD 4556, Australia

**Keywords:** vigilance, urbanization, Macropod, urban wildlife

## Abstract

**Simple Summary:**

Urban landscapes are increasing across the globe, causing wildlife to face new challenges and driving behavioural change. Wildlife in these urban landscapes must adapt their behaviour to survive. We investigated vigilance behaviour in urban and non-urban populations of eastern grey kangaroos. We found that the difference in land use alone did not affect their vigilance behaviour, but kangaroos spent more time vigilant in areas of high human population density. Season and sex also influenced the amount of time spent vigilant: more time was spent vigilant in winter and in female kangaroos. This is the first study to compare the vigilance behaviour between urban and non-urban populations of a large mammal across regions, giving the first insight into how kangaroos adapt their behaviour in urban environments.

**Abstract:**

Rapid increases in urban land use extent across the globe are creating challenges for many wildlife species. Urban landscapes present a novel environment for many species, yet our understanding of wildlife behavioural adaptations to urban environments is still poor. This study compared the vigilance behaviour of a large mammal in response to urbanisation at a landscape level. Here, we investigate urban (*n* = 12) and non-urban (*n* = 12) populations of kangaroos in two regions of Australia, and the relationship between kangaroo vigilance and urbanisation. We used a linear modelling approach to determine whether anti-predator vigilance and the number of vigilant acts performed were influenced by land use type (i.e., urban or non-urban), human population densities, kangaroo demographics, and environmental factors. Kangaroo behaviour differed between the two study regions; kangaroo vigilance was higher in urban than non-urban sites in the southern region, which also had the highest human population densities, however no effect of land use was found in the northern region. Season and sex influenced the vigilance levels across both regions, with higher levels seen in winter and female kangaroos. This study is the first to compare urban and non-urban vigilance of large mammals at a landscape level and provide novel insights into behavioural adaptations of large mammals to urban environments.

## 1. Introduction

Urbanization is increasing at a rapid rate across the globe; areas that were once undisturbed by humans are now becoming human-dominated, which increases the chance of human-wildlife interactions [1]. Wildlife remaining in these modified landscapes are forced to adapt their lifestyles due to these increasing human interactions. Habitat fragmentation and increased isolation of animal populations are common issues that are being faced as human-made barriers, such as buildings and roads, are being constructed [2]. The difference between any two habitat structures affects predation risk [3], and variation in predation risk is greater between urban and non-urban habitats than two habitats with the same land use. Predation risk to animals that are found in central, urban locations is very different to those who inhabit rural areas, with urban populations often experiencing lower predation risk when compared to rural populations [4], therefore animals may no longer need to maintain a high level of vigilance [5]. Conversely, animals that are found in urban habitats may be subjected to alternate types of predation, with a higher frequency of domesticated animals that prey on them, such as cats and dogs [6], and potential persecution by humans. This shift in predator type between urban and rural landscapes may change the anti-predator response of urban animals.

The effect that urbanisation is having on vigilance behaviour has already been studied in a number of species, with contrasting results. Any animal living within an urban environment is subject to some level of human disturbance, however the influence of the constant presence of humans on wildlife may vary between species. It could be expected that vigilance would decrease in response to the reduction in predator numbers in urban environments due to human presence. For instance, in a study conducted on red squirrels in urban environments, Uchida et al. found a decreased anti-predator response when compared to non-urban squirrels, which was likely due to habituation to human presence and consistent feeding [5]. In contrast, in a study that was conducted by Sarno et al., on eastern gray squirrels (*Sciuridae carolinensis)* animals in urban locations spent more time vigilant than ones that were found in more rural locations [7]. This was thought to be due to either increased noise level and unpredictable human disturbance, or the higher densities of squirrels that were found in urban locations, leading to increased aggression between individuals. Humans may be perceived as being a potential predator by some species. However, in urban environments, humans often do not exhibit harmful behaviour towards animals that are present. This consistent exposure for animals to human presence may lead to reduced responses to human presence [8] and a subsequent decrease in anti-predator response. In fact, the behaviour of urban animals in response to human is likely to be influenced by the types of human-wildlife interactions. In a study on eastern grey kangaroos, Austin and Ramp [9] found that kangaroos demonstrated plasticity in behavioural response to humans at fine spatial scales, yet their responses varied in the areas where human interactions were either benign or harmful.

Vigilance is a common behaviour that is displayed by prey species to increase their safety; two of its main functions are as a method of predator avoidance [10], known as anti-predator vigilance, or to gain information about other group members [11], which is known as social vigilance. The time spent vigilant can impact an individual’s ability to perform other activities, such as feeding and caring for young, thus animals must optimise this feeding/vigilance trade-off. Various factors have already been described as having an effect on vigilance, many of these are related to group dynamics, including group size [12], position in the group and distance to nearest neighbour, sex-class, reproductive status [13], and the type of environment that they are found to be living in. Anti-predator vigilance with respect to group size has been the most commonly studied form of vigilance to date; most studies show a decrease in vigilance with an increase in group size [14]. Whereas, social vigilance has been less widely studied, as it is more difficult to study due to the wide variety of roles that it includes. It covers most interactions between conspecifics, providing information on feeding, reproductive opportunities, protection of young, locating optimal feeding patches, and indirectly detecting predators. Additionally, many species are thought to have adapted vigilance behaviour so as not to incur high costs to other factors by continuing to chew or handle food whilst performing a vigilant act, which is a low-intensity form of vigilance (referred to as vigilance with chewing). The high-intensity vigilance is seen when the animal stops all other activities to perform a vigilant act (referred to as vigilance without chewing) [15]. However, carrying out multiple tasks can reduce the efficiency that each task is performed at so individuals must choose between a higher cost or less efficient behaviour. It is thought that high-intensity vigilance would more often occur in areas where the predation risk is higher.

Eastern grey kangaroo vigilance has previously been shown to be influenced by group size, distance to cover, and season, as seen reported in previous studies, such as that by Pays et al. [16], who found increasing the group size decreased the amount of time spent vigilant. Seasonal influences on kangaroo vigilance have also been found to be related to food supply fluctuations relating to seasonal variation in food supply [17], noting decreased vigilance in late summer to early autumn and that variation in food resources heavily influences group size.

We studied vigilance activity in twenty-four populations of eastern grey kangaroos, *Macropus giganteus*, across two regions of Australia: south-east Queensland (SEQ) and the Australian Capital Territory (ACT), to investigate whether urbanisation affected vigilance behaviour in kangaroos. These two regions are contrasting in overall kangaroo population sizes and densities, and also in the management of kangaroo populations. In the ACT region, kangaroos occur in high densities in both urban and non-urban environments, and they are managed according to the ACT kangaroo management plan, which covers the management of kangaroos on government and privately owned lands. This plan allows for the culling of kangaroos in conservation reserves to maintain set densities of kangaroos and for permit-based culling on private lands to mitigate property damage [18]. In contrast, SEQ populations occur in lower densities and they have been shown to be to be declining across many parts of the region [19]. There is no active management of kangaroos in the SEQ region aside from permit-based culling on private lands [20]. There are also higher levels of human population growth rates, urban expansion, and land clearing occurring in the SEQ area.

It was expected that: (i) kangaroos in urban locations would spend more time exhibiting vigilant behaviour due to increased human disturbance; and, (ii) kangaroos in the ACT would spend less time vigilant than in SEQ due to their larger mob sizes (i.e., the total population size).

## 2. Materials and Methods

### 2.1. Urban Classification

The sites were classified as either urban or non-urban based on human population density and site characteristics relevant to kangaroo ecology, such as density of buildings and other physical barriers to movement e.g., fences and the presence of remnant habitat. Human population densities were defined according to the SA2 area that each site occurred in. The SA2 areas are the smallest unit that is used to collect human population statistics by the Australian Bureau of Statistics (ABS) [21] and they are similar to a post code area. According to the ABS, urban sites were defined as having a human population density of >200 people per km^2^, while non-urban sites had densities of less than 200 people per km^2^. In this study, the term ‘land use’ is broadly used to describe an area as either urban or non-urban, however, within these definitions, there are finer scale differences in land use and the different types are outlined below.

### 2.2. Study Sites

The study was conducted across six urban and six non-urban locations in both South-East Queensland and in the Australian Capital Territory, giving a total of 24 sites. Data were collected over two study periods to account for seasonal variation: November 2015–January 2016, during summer and April–June 2016, during autumn/winter. Each site was visited once during each period, and the time of day (i.e., morning or afternoon) was kept consistent for each site between seasons. All of the recordings were taken either in the hours after dawn (0400–0900 h) or before dusk (1500–1900 h) when kangaroos are known to be most active and foraging. Predators of kangaroos in the study areas include red foxes (*Vulpes vulpes*), wedge-tailed eagles (*Aquila audax*), and occasionally domestic dogs and dingoes (*Canis lupus dingo*).

Common features of the urban sites included kangaroo proof fencing or barriers that prevented kangaroo movement, high levels of habitat fragmentation with some sites being completely isolated, and a high level of human presence or activity. The urban sites included golf courses, university campuses, memorial gardens, parks, and natural reserves within residential areas; some of the sites were in the process of further habitat disruption such as vegetation clearing or construction work, but others were stable habitats. Common features of the non-urban sites were: fewer kangaroo proof fences or barriers, a high percentage of green space or natural vegetation, no roads present or roads with a low traffic volume, full connectivity with no isolated populations, and a low presence of human activity. Non-urban sites were found on the outskirts of rural towns, within nature reserves, national parks, private wildlife reserves, campgrounds, and some were on privately owned grazing land. The vegetation present across the sites was made up of a mixture of open pasture, mixed open forest and maintained grasslands. The study areas consisted of different sized mobs of kangaroos, ranging from seven at the smallest to over 400 within national parks. A detailed list and maps of the study sites can be seen in Supporting Documents 1.

### 2.3. Recording of Behaviour

Behavioural data were collected by videotaping (using a Samsung SMX-F34 Flash Memory Camcorder, ×42 zoom) either individual subjects or a few animals. The animals filmed were on the periphery of groups if possible, however this was not possible in some locations. The recordings were taken for a maximum of 30 min. or until animals moved out of shot. Footage was taken from different locations depending on the behaviour of the animals upon arrival, some recordings were taken from the inside of a car and the rest were recorded on foot. A settling period was allowed, due to some of the animals being flighty. The time allowed for the settling period varied between sites, but observations were made once the animals had commenced normal feeding behaviour. From the footage collected, 10-minute focal observations of selected individuals were observed. A maximum of six individuals per site, consisting of three males and three females if possible, were chosen to be studied if the population size allowed. For each sample, mob size and group composition were recorded (sex of the individual was also recorded). We recorded mob size as the local population density (all kangaroos seen within 100m of focal animals), however in many cases where large mobs were present, this figure was only an estimate so mob size was categorised (0–10, 11–20, 21–30, 31–40, 41–50, 51–100, 101–150, 151–200, and over 200 individuals). Other studies often record group size according to a study that was conducted by Jarman and Coulson [22], with a maximum separation of 30 m between two neighbours, to be classified as one group; discounting juveniles. We chose to use mob size instead of group size, as the individuals were constantly changing between groups; other studies would discount the recording if the group size changed throughout the 10 min., meaning that most of our recordings of the larger groups would have to be discounted. The distance to cover of each kangaroo was measured with a Bushnell rangefinder (0–25, 26–50, 51–100, 101–200, and over 200 m). Cover was defined as vegetation that caused obstructed vision when the individual was in an upright position (open grassland) or when they were sheltering in forest.

A vigilant act was characterized as when an animal raised its head above horizontal, in either a crouched or upright position, and scanned its surroundings without moving its feet. Vigilance was further classified into either anti-predator or social vigilance, by using the individual’s head orientation to determine its visual attention. An animal was classified to be exhibiting anti-predator vigilance when it was looking away from other individuals, and social vigilance when it was looking towards other group members [23]. Vigilance with chewing was classified as when an individual was exhibiting a vigilant position whilst chewing, and vigilance without chewing was classified as when the individual was vigilant but was not chewing.

The amount of time spent in the different types of vigilance for each individual was extracted from the video recordings. For each second, the individual was either vigilant or not, and the category of vigilance being performed was recorded. From these results, the total time spent in each type of vigilance could be calculated, as well as the total amount of time spent in vigilance behaviour. We also measured the number of vigilant acts that occurred in the 10 min.; within a vigilant act an animal can exhibit more than one type of vigilance. The scan duration was measured as the length of time from the beginning of one vigilant act until a different behaviour was conducted. We calculated the total time spent in each type of vigilance, total time spent vigilant overall, and the number of vigilant events for each individual.

### 2.4. Data Analysis

Analysis was undertaken while using linear mixed models that were conducted in the R environment (R Core Team, 3.2.5, Vienna, Austria) [24]. To inform model selection and investigate which factors influenced each of the response variables: time spent vigilant and number of vigilant events, we built a conditional inference tree (R package partykit) [25] to confirm which variables should be included in the models. Two linear mixed effects models (R package lme4) [26] were constructed with total vigilance and the number of vigilant acts as the response variables and human population density (2016), region, season, land use, rainfall, distance to cover, mob size and sex as the fixed effects, and site being included as a random factor. A square root transformation was used on the total time spent displaying vigilant behaviour to satisfy the assumptions of normality; all of the results displayed are on the transformed data. As our results showed that a low proportion of time was spent exhibiting social vigilance, we did not run separate models on anti-predator and social vigilance. As the two regions varied in their methods of kangaroo management, and as we found an effect of region on the above factors, we then individually ran further models on each region to investigate if different factors were influencing vigilance within each region.

We ran a total of four linear mixed models with varying interactions to create the most optimum model possible for each variable. The final models were determined while using backwards elimination of variables based on model AICc values [27], choosing the model with the lowest AICc value as the final model. Only the factors that had an effect were left in the final model, factors that did not cause an effect are not shown. To see which fixed effects were used for each response variable in our final model, see Supporting Documents 2, Appendix A.

## 3. Results

### 3.1. Total Time Spent Vigilant

No effect of land use on total vigilance time was found overall (Est = −0.90 ± 1.63, *t* = −0.55, *p* = 0.59; Appendix A). The final model did include an interaction between season and region (Est = 4.48 ± 1.17, *t* = 3.83, *p* < 0.001; Appendix A), which suggested that time spent vigilant was higher in winter than summer and higher in ACT when compared to SEQ. More time was spent vigilant in winter for both regions. The time that kangaroos spent vigilant was lower in male kangaroos (LMM: *n* = 284 individuals (140 summer, 144 winter), Est = −1.51 ± 0.57, *t* = −2.65, *p* < 0.05; Appendix A, Figure 1). Of the total time spent vigilant, 90% was anti-predator vigilance. However, the conditional inference tree did show an effect of urban influence on total time spent vigilant and showed a significant increase in time spent vigilant when the human population density exceeded 1562.7/km2 (Figure 2).

Separate models for each region also demonstrated an effect of land use. The model for ACT suggested land use increased time spent vigilant in urban regions (Est = 3.26 ± 1.50, *t* = 2.18, *p* = 0.05; Appendix A) and also that more time was spent vigilant in winter (Est = 3.96 ± 0.83, *t* = 4.79, *p* < 0.001; Appendix A). In contrast, the final model for SEQ showed no influence of land use and suggested that time spent vigilant was lower in male kangaroos (Est = −3.46 ± 0.72, *t* = −4.80, *p* < 0.001; Appendix A) and decreased with increasing distance to cover (Est = −1.12 ± 0.44, *t* = −2.54, *p* < 0.001; Appendix A).

### 3.2. Vigilance with Chewing

Vigilance with chewing was the most common behaviour across regions and seasons, except for in urban ACT, where vigilance without chewing was highest, the higher intensity form of vigilance (Figure 3).

### 3.3. Number of Vigilant Acts

The number of vigilant acts was not found to be influenced by land use and it was only influenced by mob size, decreasing with increasing mob size in both seasons and regions (Est = −0.11 ± 0.04, *t* = −3.04, *p* < 0.05; Figure 4, Appendix A).

## 4. Discussion

Here we present the first evidence comparing urban and non-urban vigilance behaviour at a landscape level in eastern grey kangaroos. Of the two regions in our study, only urban kangaroos in the ACT spent more time vigilant than non-urban kangaroos, with no difference being seen between the urban and non-urban kangaroos in SEQ. Our results suggest that kangaroos that were found in areas of high human population density (HPD) are spending more time vigilant when compared to those found in areas of lower HPD and/or ‘non-urban’ environments. Urban sites in the ACT region had higher HPD than those in the SEQ region where no effect of land use was found. These high HPD sites are only found in this study in the ACT sites. It is noteworthy that ACT kangaroo populations, in general, have higher densities of kangaroos than SEQ and a kangaroo management plan is implemented to maintain optimum densities of kangaroos [18]. None of the sites that were used in this study had been culled in the years preceding, however two of the urban sites had been included in reproductive control trials. The linear modelling results also showed a seasonal and regional effect overall, showing that kangaroos spent more time vigilant in winter and in the ACT.

Prior to this study, the effects of urbanization on eastern grey kangaroos vigilance has only been investigated once, when comparing two populations that were found in close proximity to each other [28]. The results from our study concur with the former study, which observed a higher time being spent vigilant at the developed site as compared to the natural. The developed and natural sites that were used in their study were more similar to the sites in ACT used in our study, rather than those in SEQ. This may help us to explain why no effect was seen in SEQ, as the SEQ urban sites used tended to be smaller in both geographic area and population size, but were also more isolated populations. Whereas, our non-urban sites in SEQ tended to have smaller population sizes than ACT, and they tended to be on rural farm properties or the edge of national parks rather than being found in the middle of a national park. Our results suggest that the presence of ongoing construction is not directly influencing the amount of time that is spent vigilant, due to the lack of a difference being found between urban and non-urban kangaroos in SEQ. Rather, this could be linked to the reduction of suitable habitat and the amount of human-induced barriers at the urban sites. Similarly, Wang et al. looked at the effects of human disturbance on the vigilance levels of red-crowned cranes (*Grus japonensis*), finding that cranes invested less time foraging in more disturbed locations [29], results that concur with ours. However, they also found a group size effect unlike our findings; if the cranes were found in a larger flock size, they spent more time in vigilance behaviour.

A large number of studies have been conducted on small mammals, such as that by Mccleery, which investigated the fox squirrel’s (*Sciurus niger*) anti-predator response along an urban-rural gradient [8]. They found that squirrels have larger flight initiation distances in rural locations, as well as a higher anti-predator response to predator stimuli in these rural locations, which suggested that these animals are influenced by human presence. Burrowing owls were found to be able to recognise humans with domesticated dogs as having a higher threat level than humans alone [30]. It is noteworthy that the sites in our study that were found to have the highest amount of time spent vigilant were those that we had observed a high number of dog walkers. The presence of domestic animals was not considered in our models, however this could be useful to investigate in future studies. The results from these studies on these other species suggest that urbanization does have an effect on the vigilance behaviours within species, however the relationship is complex and it varies between taxa. As we found differences in the effect on vigilance behaviour between the urban sites between regions, we suggest further research be carried out at more sites over urban-rural gradients.

Vigilance with chewing is thought to be a lower intensity form of vigilance than when individuals are not chewing as the animals are performing two activities at once. In this study, kangaroos mostly followed similar patterns that were seen in other studies on both eastern grey kangaroos [14] and other herbivores, such as zebra [31], with a higher level of vigilance with chewing being observed. However, we note a higher level of vigilance without chewing was seen in the urban locations in the ACT, which suggested that these kangaroos might be more alert to predation. Although there appeared to be a difference in vigilance with and without chewing, it should be noted that we did not include this in our models, as the aim of our study was not to differentiate between the different types of vigilance. The urban ACT sites were all found in the centre of residential areas, with a high level of dog walkers present; although, it is uncertain from this study whether this is the cause of the higher level of vigilance without chewing, it is thought to be a potential contributing factor. It was also noted on some of the days that the kangaroos were observed in the urban ACT sites that there was a particularly high wind speed. High wind speed has been seen to affect the vigilance levels in previous studies of eastern grey kangaroos and other macropod species, such as the brush-tailed rock wallaby (*Petrogale penicillata*) [32]. Overall, vigilance scans with chewing were more frequently used than without chewing, as expected, this is also observed in other grazing herbivores, such as female bison (*Bison bison*) and elk (*Cervus canadensis*) in [33], fitting the hypothesis that these foragers try to limit their vigilance to have as little cost to their foraging time as possible.

Urban ACT kangaroo populations were noted to spend the highest amount of time vigilant, including without chewing, the higher-intensity form of vigilance. This observation is in contrast to what we expected to find; based on previous research, we anticipated SEQ urban kangaroos to have the highest vigilance and non-urban ACT to have the lowest vigilance. Given that predators (natural or domestic animals) were seen in some form at both urban and non-urban sites, the differences that were seen in our study may also possibly be linked to the differences in nature of the urban sites and kangaroo management between the two regions. It is possible that the differences in vigilance behaviour between urban environments may also be due to the nature of human activity and human-kangaroo interaction in the sites, as investigated in a recent study by Austin and Ramp [9]. They found that eastern grey kangaroos exhibited behavioural plasticity at a fine scale in response to different levels of human interactions.

We suggest that the pattern seen here could also be associated with the different types of encounters that kangaroos have with humans. In three of the ACT sites this could potentially be linked to active kangaroo management; over time if kangaroos are experiencing negative encounters with humans, such as when they are tagged or when reproductive controls are used, this may make the kangaroos more wary in these human-dominated habitats. However, human interactions at the other three ACT sites were non-antagonistic. Therefore, we suggest further study on urban kangaroo populations subjected to infrastructure development and habitat fragmentation, as well as across a variety of urban land use types to investigate the ability of kangaroos to adapt their behaviour within changing urban environments. This may also provide further insight into why some urban kangaroo populations are stable or increasing (as in the ACT) while others are decreasing (SEQ) in these urban landscapes.

Our results found no effect of human population density on the frequency of vigilant acts. The only influencing factor found was mob size, which is in agreement to other studies that show eastern grey kangaroos exhibit a ‘group-size effect’ [34], and also similar to many other prey species [9]. It is thought that it is easier for larger groups to detect a predator due to the increased numbers of individuals that can survey their surroundings (‘detection effect’ hypothesis [35]). Increasing group size can also have the opposite effect, with less time being spent vigilant, as there is now an increase in intra-specific competition for food, which causes the individuals to invest more time feeding (‘scramble competition’ hypothesis [36,37]). In a study that was conducted by Carter et al., individual differences were noted with some not showing a group size effect at all, additionally they noted a difference between female kangaroos in different reproductive states [38]. This was also seen in a study that combines predation risk with group size effects, females with young became more solitary as predation risk decreased [39]. Our results also showed that, although kangaroos in the ACT were exhibiting fewer vigilant acts than those in SEQ, they spent more time vigilant overall, which suggested they are spending more time pausing other activities and scanning when compared to the SEQ kangaroos. This could also be linked to them becoming more wary due to any negative interactions with humans. We chose not to include the reproductive state of the females as reproductive state could not always reliably be ascertained when observing kangaroos from a distance. The kangaroos we studied were seen to follow either the ‘detection effect’ hypothesis [35] or the ‘dilution effect’ [40], that the larger the group, the smaller the risk of predation due to an increase in the number of animals that are vigilant.

Distance to cover was also seen to have an influence on the amount of time spent vigilant in SEQ. The amount of time spent vigilant decreased as the animals were found farther from cover. It is known that the effect distance to cover has can vary widely in prey species, with it being non-existent in some cases [41,42]. Cover is an important factor for eastern grey kangaroos as: they use it to rest in, will flee to it when they are alarmed, and spend most of their time feeding close to cover when they are considered in danger. Some studies conducted on the eastern grey kangaroos have shown no relationship between distance to cover and vigilance, and this is thought to be due to the different hunting strategies that are used by eastern grey kangaroo predators [43]. One of the predation risks for eastern grey kangaroos is from avian predators; raptors gain most success when the kangaroos are found out in the open where as their terrestrial predators have more success when the animals are near to cover, as it is easier for them to sneak up and ambush them [23]. As our results show a decrease in vigilance as the distance to cover increased, but a higher amount of time spent vigilant in winter, we suggest the main predation risk to the animals that we studied is from terrestrial predators rather than avian species. If avian predation was their main threat, we would have expected their time spent vigilant to increase with increasing distance to cover. Additionally, the higher amount of time spent vigilant in winter could easily be explained by a reduction in flora cover due to the seasonal fluctuations.

## 5. Conclusions

To conclude, our results suggest that, although land use type alone does not influence the vigilance levels in eastern grey kangaroos, those found in areas of higher human population density spent more time vigilant. We also confirm the findings of other studies, that a combination of different factors, not just a single factor, all influence the kangaroo’s decisions to invest in vigilance behaviour. Finally, we highlight that our results suggest that the nature of the urban environment may influence kangaroo vigilance behaviour, which is higher in some urban environments, as our study compared kangaroo populations that are rapidly declining in changing urban environments [19] to those that are thriving. We suggest further research comparing the populations that are being subjected to new development with those in stable environments, and also along urban-rural gradients to improve our understanding of kangaroos’ adaptations of vigilance behaviours over time.

## Figures and Tables

**Figure 1 animals-09-00494-f001:**
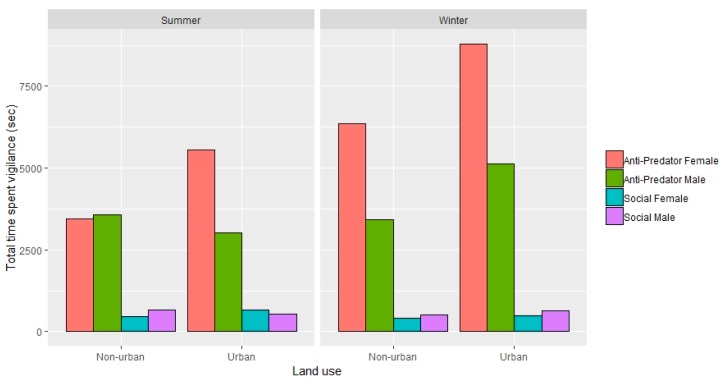
Total time spent displaying anti-predator or social vigilance in eastern grey kangaroos across two regions (south-east Queensland and the Australian Capital Territory).

**Figure 2 animals-09-00494-f002:**
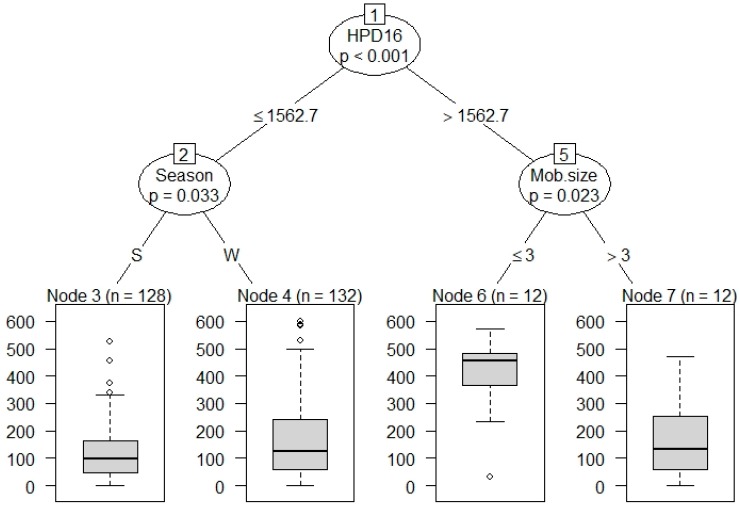
Conditional inference tree showing variables predicting the total time spent vigilant (sec) during ten-minute intervals (*y*-axis) by eastern grey kangaroos (HPD16—human population densities (persons/km^2^) as of 2016).

**Figure 3 animals-09-00494-f003:**
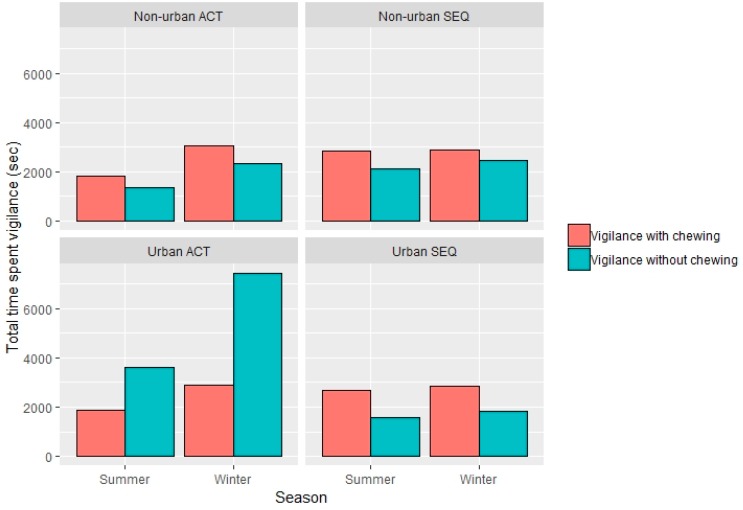
Total time spent displaying vigilance with or without chewing in eastern grey kangaroos found in different types of land use in south-east Queensland and the Australian Capital Territory.

**Figure 4 animals-09-00494-f004:**
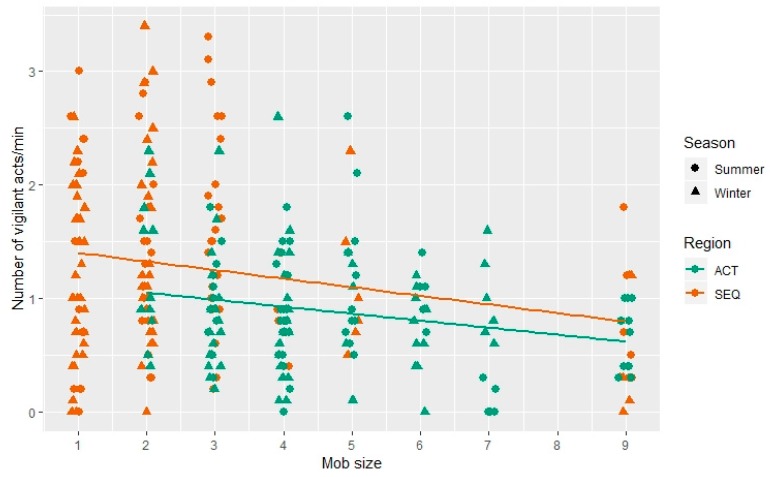
Frequency of vigilant acts per minute in relation to mob size of eastern grey kangaroos grazing in two different regions; south-east Queensland (SEQ) and the Australian Capital Territory (ACT) (*n* = 284 individuals). Mob size is defined by categories: 1 (0–10), 2 (11–20), 3 (21–30), 4 (31–40), 5 (41–50), 6 (51–100), 7 (101–150), 8 (151–200), and 9 (over 200 individuals).

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
