# Peer review of "Eastern Grey Kangaroo (*Macropus giganteus*) Vigilance Behaviour Varies between Human-Modified and Natural Environments"

_animals, 2019, doi:10.3390/ani9080494_

Round 1

Reviewer 1 Report

General comments

This paper addresses vigilance behavior comparison between in urban and non-urban populations of eastern grey kangaroos at a landscape level, and suggests that although land use type alone does not influence their valiance levels, those found in areas of higher human population density spent more time vigilant. The data are potentially interesting and worthy of eventual publication.  However, the writing lacks clarity and sharpness in introduction and discussion section. Pleas add more detail explanations for three predictions to investigate if urbanization is having an effect vigilance behavior in kangaroo. Discussion should be focusing the results of predictions.

Specific comments

Introduction

Please explain difference between group size and mob size, which is not familiar word for me.  (L334)

L80 delete “it covers” which repeated twice.

L90  anthropogenic disturbance. This should be clarified.

2. Materials and Methods

2.1 Study Site

L104-117 In addition to the common feature of the urban and non-urban sites, it is necessary to describe comparison of mob size and degree in anthropogenic disturbance (HPD?) between SEQ and ACT in Supporting Documents 1, because these features were used for prediction ii) and iii).

L120. Please explain in detail on “site characteristics relevant to kangaroo ecology”. 

Discussion

In discussion sections, it is better insert subhead line corresponding to Results as 4.1 Total time spent vigilant, 4.2 Vigilance with chewing, and 4.3 Number of vigilant acts. 

L228-232: Data on HPD and densities of kangaroos in study sites should be presented in Supporting Documents, because these are important factors effect on time vigilant of kangaroos.

L266-267 Why are urban kangaroo populations stable or increasing in the ACT and others decreasing in SEQ?  Are there any differences in some reasons such as habitat quality, kangaroo population history and management approach between the two regions?

L344  I am not familiar with the differences between mob size and group size,

Please explain it.

Fig S1  

The study area is not well known to many of the readers of Journal of Animals. It may be an idea to show an additional topographical map of the area, including the locations of the major settlements.

Author Response

Response to Reviewer 1 Comments

Point 1: Please explain difference between group size and mob size, which is not familiar word for me.  (L334)

Response 1: methods section has been updated to clarify definitions of mob size and references to group size have been reduced as mob size is used in modelling, explained in lines 169-177.

Point 2: L80 delete “it covers” which repeated twice.

Response 2: I have changed the first term to it includes

Point 3: L90 anthropogenic disturbance. This should be clarified.

Response 3: Some examples have now been given

Point 4: L104-117 In addition to the common feature of the urban and non-urban sites, it is necessary to describe comparison of mob size and degree in anthropogenic disturbance (HPD?) between SEQ and ACT in Supporting Documents 1, because these features were used for prediction ii) and iii).

Response 4: I have added these figures to the table in Supporting Documents 1.

Point 5: L120. Please explain in detail on “site characteristics relevant to kangaroo ecology”.

Response 5: This has been explained in further detail.

Point 6: In discussion sections, it is better insert subhead line corresponding to Results as 4.1 Total time spent vigilant, 4.2 Vigilance with chewing, and 4.3 Number of vigilant acts.

Response 6: I feel this breaks up the flow of the discussion and is something that is not often used in the discussion.

Point 7: L228-232: Data on HPD and densities of kangaroos in study sites should be presented in Supporting Documents, because these are important factors effect on time vigilant of kangaroos.

Response 7: This data has been added to the table in supporting documents 1

Point 8: L266-267 Why are urban kangaroo populations stable or increasing in the ACT and others decreasing in SEQ?  Are there any differences in some reasons such as habitat quality, kangaroo population history and management approach between the two regions?

Response 8: ACT has a well implemented kangaroo management plan which has been mentioned in other parts of the discussion.

Point 9: L344 I am not familiar with the differences between mob size and group size, Please explain it.

Response 9: as explained in Response 1

Point 10: Fig S1 The study area is not well known to many of the readers of Journal of Animals. It may be an idea to show an additional topographical map of the area, including the locations of the major settlements.

Response 10: A replacement map has been produced

Reviewer 2 Report

This study by Hume et al. investigates factors influencing vigilance behavior in Eastern Grey Kangaroos. Their goal is to assess the influence of human presence (via human density) and urbanization, a topic apt for research in an increasingly human-modified world. The study is simple and has potential, but lacks focus, including predictions justified by evidence-based rationale, a clear connection between predictions, methods, and results, and a discussion that places the results into a broader biological context. There are also several grammatical errors, run-on sentences, and typos. The manuscript will require more editing before being ready for publication.

Main concerns:

1)      The introduction provides reasonable rationale for the study (i.e., why the study is needed), however, fails to delivery on justifying the predictions. This study looks at and finds effects of season, region, sex, human density, and group size. What were the predictions for each of these factors? In particular, the inclusion of region needs to be justified with background information and evidence to support clear predictions. I have more detailed comments below.

2)      The statistical analyses are confusing and seem overly complicated for such a simple study. The authors should just include into a single linear mixed model the factors for which they have valid rationale for including (those highlighted in the introduction as being important in influencing vigilance). Their single model finds a region by season interaction and so they ran two separate models per region to evaluate this interaction. This is all that needs to be done. There is not a strong justification for running multiple models and performing model selection if there are clear predictions on a limited number of variables (which this study seems to, although the predictions need to be better justified). If the authors feel that their methods are appropriate, this should be better explained. Also, there are several inconsistencies between the methods section and the results (detailed below) so perhaps this issue could be resolved through better integration between these two sections.

3)      The Discussion needs a lot of work to go beyond comparing the results to other studies (e.g., this study found the same results as us or this study found different results). The Discussion should tie together the independent results of the study into a concise message on what the authors think is happening in their system and then use external references to support their claims. For example, the majority of the significant results seem to be driven by the urban ACT sites, which show almost opposite patterns as all other sites. The discussion could focus on what makes these areas more different than the others. The authors briefly state that these areas have kangaroo management because of higher kangaroo densities in the urban ACT sites. This alone might tie together all the results, but is only mentioned in a single passing sentence.

Minor comments/edits:

Abstract:

Lines 16-17: is there no comparable study of deer in urban environments?

Introduction:

Line 41 and elsewhere: change “man-made” to human-made

Line 52: remove apostrophe from squirrel’s

Line 58: add areas or habitats after the word urban

Line 66: remove the word “on” after impact

Line 86: insert the word “on” between effect and vigilance

Lines 72-82: Paragraph is awkward, disorganized and lacks focus.

Predictions need more justification. See below.

-          Prediction 1: Why would you expect kangaroos to be more vigilant in urban areas? You explain conflicting results from previous research (the squirrel studies) in the Introduction and explain how predators tend to be lower in urban areas besides cats and dogs. It seems that kangaroos are likely too big to be consistently preyed upon by domestic pets so they should technically experience lower predation risk in urban habitats leading to lower levels of vigilance (why waste time looking for predators that are not there?). Please explain the logic behind your reasoning.

-          Prediction 2: what is a mob size? Does this refer to group size? Why are group sizes larger in the ACT?

-          Prediction 3: same issue as prediction 1. Need to explain why anthropogenic disturbance should lead to higher vigilance if human presence lowers predation risk. Also, if SEQ has both higher anthropogenic disturbance and smaller group sizes compared to ACT, how can you tell which factor is more strongly influencing vigilance behavior? How will you decouple these factors?

Overall, more information is needed on the different regions (ACT an SEQ). Why were they selected and how do they differ from each other?

Methods:

Line 96: you can’t really make inferences on seasonal effects if you have a sample size of one per season. I would downplay this in your manuscript.

Line 188: Urban classification should come before study site sub-section. So you used human population density to quantify urban vs. non-urban and then found 12 sites within each region that met these definitions even though structurally they are quite different.

Line 123: what does ABS stand for?

Line 134: how did you define normal behaviors? Why did you not standardize the “settling” period. This just introduces another source of error into your results

Line 138: how was kangaroo sex determined?

Results:

Several inconsistencies between results and the methods:

-          Distance from observer was measured, but then never analyzed. Did this affect results in any way?

-          Separate models for each region are presented in the Results, but the rationale for breaking these into two different models is not explained in the Methods.

-          Vigilance while chewing vs not chewing is not explained in the methods and is then presented in the results as a response variable that was examined. Was a statistical analysis performed on this variable or are these just descriptive results?

-          Categorical variable urban vs non-urban is not included in the statistical models (instead a continuous variable, human population density, is included in the model) but is then presented in results.

-          Number of vigilant acts as a dependent variable in a model when only total time spent vigilant is explained in the methods as being the dependent variable.

-          Mob size category assignments need to be presented earlier in the ms as the presentation of results conflict with how mob size is categorized in the methods.

-          Aspects of the models presented in supplementary material are not explained in the manuscript, but then their results are discussed and interpreted. Examples: inclusion of rainfall as a predictor variable, inclusion of human density AND land use (urban vs non-urban) as predictor variables (only human density is explained in the manuscript), model selection using AIC values (are these from the conditional inference trees?). The model selection approach is confusing and more explanation is needed for clarity.

Line 184 and elsewhere: I do not think the term “land use” is appropriate for your study. You define urban vs. non-urban based on human population density not land use. Indeed, your urban sites vary widely in land use (some are golf courses, some universities, some are even natural reserves!). Please change this term throughout and be specific on what variable you are actually evaluating.

Figure 2 need more explanation as conditional inference trees might not be familiar to many readers. What is presented on the y-axis (it’s different from figure 1)? Define what HPD16 means at the top.

Line 185: Figure 1 is cited here, but this figure does not show an interaction between season and region.

Discussion:

Lines 230-231: how does HPD of non-urban sites in ACT compare to those in SEQ? Same or also higher?

Lines 232-234: This is especially noteworthy and needs to be brought up in the Introduction. If kangaroos experience negative encounters with humans in ACT it makes sense that they are more wary in human-dominated habitats. Is this a hunting vs non-hunting situation? This has huge implications for both the rationale of the study and interpretation of study’s results (e.g., is hunting seasonal? Are certain sexes targeted?). This explains why your results might be in contrast to other studies (which are detailed in the Discussion).

Line 242: awkward sentence. Please re-phrase.

Line 258: grammar/typos: “…modelling, however, this could be useful…”

Lines 266-267: Again, this information would have been useful information in the Introduction to help justify your study and the rationale behind your predictions.

Lines 269-278: What is the point of this information? If social vigilance has been shown to be negligible in kangaroos in past studies, why was it even included in this study? No need to explain why you included both sexes instead of just females. This just weakens the robustness of your study design.

Line 279: …as expected.  Predictions for how females and males might respond were not explained in the Introduction or Methods.

Lines 280-281: Cannot make this claim that males in your study had higher levels of social vigilance than females. This was not evaluated statistically and the plots show nearly equal values with females. Also, it’s not clear why there is so much focus on social vigilance when, as stated above, this response has been shown to be negligible in kangaroos.

Line 282: Rephrase “there has been seen to be…”  Also, this is a run-on sentence.

Line 295-296: what is meant by “more commonly observed”? First, this is not a grammatically correct sentence and what is more commonly observed and based on what evidence? I see this is explained more further down. Please rewrite this introductory sentence as it is confusing.

Line 306: delete “seen in” or “observed in” (typos)

Lines 306-309: these final sentences provide the same information as on lines 296-298. Some re-organizing of ideas within this paragraph could be helpful. Also, all interpretations within this paragraph are speculative and are not supported by the results. If humans actively manage kangaroos in the urban ACT sites, could this potentially explain the higher incidence of vigilance without chewing?

Lines 315-316: delete “which is more commonly seen.”

Line 316: Delete “cause”

Line 324: Contradictory to how the urban sites are presented in the Introduction as having “kangaroo proof fencing or barriers which prevented kangaroo movement, high levels of habitat fragmentation with some sites being completely isolated.”

Lines 322-328: Missed opportunity to discuss the unusual patterns of vigilance of kangaroos in the ACT. Your results show that they are more vigilant in terms of time spent vigilant (in urban areas), but actually exhibit fewer vigilance acts than kangaroos in SEQ. This suggests that they are spending more time per act (longer periods of pausing and looking) compared to SEQ kangaroos.

Lines 330-338: This info belongs in the Methods section as justification for your methods.

Line 344: insert “on” between the words influence and the

Line 341: change further to farther (farther refers to distance)

Lines 351-355: These interpretations are not logical. Why would perceived risk decrease with distance from cover if kangaroos are being hunted by terrestrial predators? Would they still not hide in vegetation? If not, you should explain which antipredator behaviors they exhibit in response to these different predators. The last sentence of this paragraph contradicts everything that comes before. You state that vigilance may be higher in winter because of reduced cover, but previously hint at cover not being important because the kangaroos are likely not being hunting by aerial predators.

Interpretations on seasonal effects should be limited as this study was conducted during only one winter and one summer. How do we know these seasonal effects persist throughout the years?

Author Response

Response to reviewers 2 comments

1)      The introduction provides reasonable rationale for the study (i.e., why the study is needed), however, fails to delivery on justifying the predictions. This study looks at and finds effects of season, region, sex, human density, and group size. What were the predictions for each of these factors? In particular, the inclusion of region needs to be justified with background information and evidence to support clear predictions. I have more detailed comments below.

Response: We have rewritten and restructured the introduction

2)      The statistical analyses are confusing and seem overly complicated for such a simple study. The authors should just include into a single linear mixed model the factors for which they have valid rationale for including (those highlighted in the introduction as being important in influencing vigilance). Their single model finds a region by season interaction and so they ran two separate models per region to evaluate this interaction. This is all that needs to be done. There is not a strong justification for running multiple models and performing model selection if there are clear predictions on a limited number of variables (which this study seems to, although the predictions need to be better justified). If the authors feel that their methods are appropriate, this should be better explained. Also, there are several inconsistencies between the methods section and the results (detailed below) so perhaps this issue could be resolved through better integration between these two sections.

Response: We only ran two models – total time spent vigilant and the number of vigilant acts. We then ran models for each region investigating total time spent vigilant. There are two tables for each model in the supplementary section to explain which the final model for either total time spent vigilant or the number of vigilant acts.

3)      The Discussion needs a lot of work to go beyond comparing the results to other studies (e.g., this study found the same results as us or this study found different results). The Discussion should tie together the independent results of the study into a concise message on what the authors think is happening in their system and then use external references to support their claims. For example, the majority of the significant results seem to be driven by the urban ACT sites, which show almost opposite patterns as all other sites. The discussion could focus on what makes these areas more different than the others. The authors briefly state that these areas have kangaroo management because of higher kangaroo densities in the urban ACT sites. This alone might tie together all the results, but is only mentioned in a single passing sentence.

Response: The discussion has now been rewritten and ties together the influence of the ACT kangaroo management plan and our results

Abstract:

Point 1: Lines 16-17: is there no comparable study of deer in urban environments?

Response 1: The authors are not aware of any studies assessing behaviour across multiple populations and regions in relation to urban impacts

Introduction:

Point 2: Line 41 and elsewhere: change “man-made” to human-made

Response 2: Done

Point 3: Line 52: remove apostrophe from squirrel’s

Response 3: Done

Point 4: Line 58: add areas or habitats after the word urban

Response 4: “Habitats” added

Point 5: Line 66: remove the word “on” after impact

Response 5: Removed

Point 6: Line 86: insert the word “on” between effect and vigilance

Response 6: Added word “on”

Point 7: Lines 72-82: Paragraph is awkward, disorganized and lacks focus.

Response 7: I have rephrased this paragraph

Predictions need more justification. See below.

Point 8: Prediction 1: Why would you expect kangaroos to be more vigilant in urban areas? You explain conflicting results from previous research (the squirrel studies) in the Introduction and explain how predators tend to be lower in urban areas besides cats and dogs. It seems that kangaroos are likely too big to be consistently preyed upon by domestic pets so they should technically experience lower predation risk in urban habitats leading to lower levels of vigilance (why waste time looking for predators that are not there?). Please explain the logic behind your reasoning.

Response 8: Urban kangaroos have only recently become subjected to human presence, therefore we would not expect them to be habituated to human presence yet. Also, kangaroos perceive dingos to be one of their biggest predators to their young, therefore having a high presence of domesticated dogs they could potentially perceive as a predator towards their young. We have amended text in the introduction to help explain the hypotheses

Point 9: Prediction 2: what is a mob size? Does this refer to group size? Why are group sizes larger in the ACT?

Response 9: Text has been amended to clarify this and we have explained the difference between group size and mob size in lines 165-169

Point 10: Prediction 3: same issue as prediction 1. Need to explain why anthropogenic disturbance should lead to higher vigilance if human presence lowers predation risk. Also, if SEQ has both higher anthropogenic disturbance and smaller group sizes compared to ACT, how can you tell which factor is more strongly influencing vigilance behavior? How will you decouple these factors?

Response 10: Text has been added in the introduction to give background on these hypotheses.

Point 11: Overall, more information is needed on the different regions (ACT and SEQ). Why were they selected and how do they differ from each other?

Response 11: I have added in more information about the regions in lines 87-89.

Methods:

Point 12: Line 96: you can’t really make inferences on seasonal effects if you have a sample size of one per season. I would downplay this in your manuscript.

Response 12: Text has been amended, data were collected over two seasons as this has been identified as a factor that influences vigilance.

Point 13: Line 188: Urban classification should come before study site sub-section. So you used human population density to quantify urban vs. non-urban and then found 12 sites within each region that met these definitions even though structurally they are quite different.

Response 13: I have rearranged these two sections. Text has been modified to improve clarity. Sites were chosen first on the basis of the definition of urban based on human population densities. While the urban sites may have varied in terms of their actual land use, they all had factors generally associated with urban areas as noted in the methods section (lines 150-155)

Point 14: Line 123: what does ABS stand for?

Response 14: It is the abbreviation for Australian Bureau of Statistics, I have now clarified this

Point 15: Line 134: how did you define normal behaviors? Why did you not standardize the “settling” period. This just introduces another source of error into your results

Response 15: Text has been amended to clarify. Normal behaviour was classified as when the animal commenced feeding as we deemed this meant they were no longer influenced by our presence.

Point 16: Line 138: how was kangaroo sex determined?

Response 16: This has now been clarified (line 185)

Results:

Several inconsistencies between results and the methods:

Point 17: Distance from observer was measured, but then never analyzed. Did this affect results in any way?

Response 17: Exploratory models were run initially and this was included as a factor but was always one of the first to be removed from the models as it was non-significant, it was also not influential in the conditional inference tree. Therefore we have removed it from the methods.

Point 18: Separate models for each region are presented in the Results, but the rationale for breaking these into two different models is not explained in the Methods.

Response 18: The rationale behind this has been explained in lines 207-210.

Point 19: Vigilance while chewing vs not chewing is not explained in the methods and is then presented in the results as a response variable that was examined. Was a statistical analysis performed on this variable or are these just descriptive results?

Response 19: I have now added this in the methods and these are only descriptive results

Point 20: Categorical variable urban vs non-urban is not included in the statistical models (instead a continuous variable, human population density, is included in the model) but is then presented in results.

Response 20: This was included in the original models, I had missed it out of the list in the methods, it has now been added.

Point 21: Number of vigilant acts as a dependent variable in a model when only total time spent vigilant is explained in the methods as being the dependent variable.

Response 21: I have now added this in the methods

Point 22: Mob size category assignments need to be presented earlier in the ms as the presentation of results conflict with how mob size is categorized in the methods.

Response 22: I have now clarified this throughout the report

Point 23: Aspects of the models presented in supplementary material are not explained in the manuscript, but then their results are discussed and interpreted. Examples: inclusion of rainfall as a predictor variable, inclusion of human density AND land use (urban vs non-urban) as predictor variables (only human density is explained in the manuscript), model selection using AIC values (are these from the conditional inference trees?). The model selection approach is confusing and more explanation is needed for clarity.

Response 23: This has now been clarified in methods L188-190 and all variables have been stated in L177-178

Point 24: Line 184 and elsewhere: I do not think the term “land use” is appropriate for your study. You define urban vs. non-urban based on human population density not land use. Indeed, your urban sites vary widely in land use (some are golf courses, some universities, some are even natural reserves!). Please change this term throughout and be specific on what variable you are actually evaluating.

Response 24: We considered the reviewers perspective on this however have kept the term land use throughout the study as we could not identify an alternative term that did not over complicate the manuscript. We have added text in the methods to clarify our definition of land use specific to this study

Point 25: Figure 2 need more explanation as conditional inference trees might not be familiar to many readers. What is presented on the y-axis (it’s different from figure 1)? Define what HPD16 means at the top.

Response 25: Figure caption has been amended to clarify

Point 26: Line 185: Figure 1 is cited here, but this figure does not show an interaction between season and region.

Response 26: Reference to figure caption removed

Discussion:

Point 27: Lines 230-231: how does HPD of non-urban sites in ACT compare to those in SEQ? Same or also higher?

Response 27: HPD of non-urban sites is now listed in supporting doc 1.

Point 28: Lines 232-234: This is especially noteworthy and needs to be brought up in the Introduction. If kangaroos experience negative encounters with humans in ACT it makes sense that they are more wary in human-dominated habitats. Is this a hunting vs non-hunting situation? This has huge implications for both the rationale of the study and interpretation of study’s results (e.g., is hunting seasonal? Are certain sexes targeted?). This explains why your results might be in contrast to other studies (which are detailed in the Discussion).

Response 28: While some kangaroo populations are actively managed through culling, none of the urban sites used in this study had been culled previously. Text has been added in discussion to clarify this (line 296-297)

Point 29: Line 242: awkward sentence. Please re-phrase.

Response 29: I have rephrased this

Point 30: Line 258: grammar/typos: “…modelling, however, this could be useful…”

Response 30: I have also changed this

Point 31: Lines 266-267: Again, this information would have been useful information in the Introduction to help justify your study and the rationale behind your predictions.

Response 31: This has now been added in the intro in lines 87-88

Point 32: Lines 269-278: What is the point of this information? If social vigilance has been shown to be negligible in kangaroos in past studies, why was it even included in this study? No need to explain why you included both sexes instead of just females. This just weakens the robustness of your study design.

Response 32: I have removed this paragraph.

Point 33: Line 279: …as expected.  Predictions for how females and males might respond were not explained in the Introduction or Methods.

Response 33: Deleted

Point 34: Lines 280-281: Cannot make this claim that males in your study had higher levels of social vigilance than females. This was not evaluated statistically and the plots show nearly equal values with females. Also, it’s not clear why there is so much focus on social vigilance when, as stated above, this response has been shown to be negligible in kangaroos.

Response 34: I have now removed this paragraph as it is not relevant to the key focus of the paper.

Point 35: Line 282: Rephrase “there has been seen to be…”  Also, this is a run-on sentence.

Response 35: This has been rephrased

Point 36: Line 295-296: what is meant by “more commonly observed”? First, this is not a grammatically correct sentence and what is more commonly observed and based on what evidence? I see this is explained more further down. Please rewrite this introductory sentence as it is confusing.

Response 36: I have rephrased this

Point 37: Line 306: delete “seen in” or “observed in” (typos)

Response 37: I have deleted “seen in”

Point 38: Lines 306-309: these final sentences provide the same information as on lines 296-298. Some re-organizing of ideas within this paragraph could be helpful. Also, all interpretations within this paragraph are speculative and are not supported by the results. If humans actively manage kangaroos in the urban ACT sites, could this potentially explain the higher incidence of vigilance without chewing?

Response 38: The kangaroo management plan is implemented to maintain the fauna in ACT as the kangaroos are one of their key species. It is not their to actively control the kangaroos but to maintain a healthy population and allow them to have corridors throughout the city.

Point 39: Lines 315-316: delete “which is more commonly seen.”

Response 39: Deleted

Point 40: Line 316: Delete “cause”

Response 40: Deleted

Point 41: Line 324: Contradictory to how the urban sites are presented in the Introduction as having “kangaroo proof fencing or barriers which prevented kangaroo movement, high levels of habitat fragmentation with some sites being completely isolated.”

Response 41: This only stated common features of the sites, kangaroos had often dug holes under the fences, and there was variation in the levels of habitat fragmentation

Point 42: Lines 322-328: Missed opportunity to discuss the unusual patterns of vigilance of kangaroos in the ACT. Your results show that they are more vigilant in terms of time spent vigilant (in urban areas), but actually exhibit fewer vigilance acts than kangaroos in SEQ. This suggests that they are spending more time per act (longer periods of pausing and looking) compared to SEQ kangaroos.

Response 43: This has now been discussed.

Point 44: Lines 330-338: This info belongs in the Methods section as justification for your methods.

Response 44: This has been changed to the Methods

Point 45: Line 344: insert “on” between the words influence and the

Response 45: This has been added

Point 46: Line 341: change further to farther (farther refers to distance)

Response 46: This has been done

Point 47: Lines 351-355: These interpretations are not logical. Why would perceived risk decrease with distance from cover if kangaroos are being hunted by terrestrial predators? Would they still not hide in vegetation? If not, you should explain which antipredator behaviors they exhibit in response to these different predators. The last sentence of this paragraph contradicts everything that comes before. You state that vigilance may be higher in winter because of reduced cover, but previously hint at cover not being important because the kangaroos are likely not being hunting by aerial predators.

Response 47: I have removed this paragraph as I do not think it was relevant to the key focus of the paper

Point 48: Interpretations on seasonal effects should be limited as this study was conducted during only one winter and one summer. How do we know these seasonal effects persist throughout the years?

Response 48: The discussion has been rewritten and with less emphasis on seasonal effects.

Round 2

Reviewer 1 Report

 The revised paper was greatly improved in terms of the readability. I have one minor comment. I am interested in the differences in vigilance behaviour may be linked to the differences in kangaroo management of the two regions. Hunting and/or culling thorough management plan may influence the vigilance behaviour. The autrous explained ACT kangaroo populations have been managed to maintain optimum density, however, did not explain management approach for SEQ kangaroo populations. Please explain more detail the difference management between two regions. 

Author Response

Point 1: The revised paper was greatly improved in terms of the readability. I have one minor comment. I am interested in the differences in vigilance behaviour may be linked to the differences in kangaroo management of the two regions. Hunting and/or culling thorough management plan may influence the vigilance behaviour. The authors explained ACT kangaroo populations have been managed to maintain optimum density, however, did not explain management approach for SEQ kangaroo populations. Please explain more detail the difference management between two regions.

Response 1: Text has been amended to clarify in the introduction - line 102

Reviewer 2 Report

I find that the revised version is much improved; the writing is more clear and the methods and results are now easier to understand. The additional information regarding the study regions has helped immensely to understand the results. The Discussion has also been modified to summarize and interpret the findings in light of previous research. I am still a bit confused by some of the statistical methods and I have a few more minor comments detailed below.

Line 109: how specifically did group size influence the amount of time spent vigilant? Positive or negative?

Line 111: Same as above, what specific effect did food resources have?

Lines 111-113: This sentence is awkwardly placed before the main predictions. Consider rephrasing.

Lines 113-116: Based on these predictions urban SEQ should exhibit the highest vigilance and non-urban ACT should exhibit the lowest. This could be expanded upon in the Discussion as the results do not support these predictions and this could be due to a variety of factors (as were mentioned by the authors). In sum, it’s interesting that the results deviate from the predctions.

Line 125: Add a comma between ABS and urban

Lines 136-137: what is meant by both observations? It says each site was visited only once.

Line 157: Change “behavioural data was” to behavioural data were

Lines 180-184: The info in these sentences was already written above. Please delete.

Line 188: add apostrophe to individuals head

Line 199: I’m confused by the inclusion of scanning rate as this is not mentioned further in the analyses or results. Is this synonymous with vigilant events?

Data Analysis section: Still a bit confusing. My interpretation is that the conditional inference tree was used to identify the variables to include in the linear mixed models. These variables were human population density, region, season, land use, rainfall, distance to cover, mob size and sex (as indicated on lines 211-212), and then model selection was done using AICc values. Is this correct? If not, then this section should be revised to explain.

Lines 215-216: Here the authors explain why they did not run a model on social vigilance (which makes sense), but do not explain why no models were used to evaluate vigilance while chewing verses not chewing. These behaviors were quantified and are displayed in the results section, but there is no explanation for why a statistical test was not performed to assess factors affecting these behaviors.

Lines 216-219: It’s not clear why this method was done as opposed to including a region*land use type interaction into the main model.

Line 221: says a total of four models were run, but the supplementary show models 0-5.

Lines 233-234: seems to be editing errors here: “(LMM: n=284 233 individuals (140 summer…”

Figure 2 y-axis: how is this in the hundreds, but in the thousands in fig 1 and fig 3?

Line 254: typo: (persons/km2) as at 2016).  Also parentheses inside of parentheses.

Lines 304-306: Awkward sentence. Please re-phrase.

Line 333: “However, a higher level of vigilance without chewing was found in the urban locations in the ACT…” This was not evaluated statistically and should be acknowledged.

Line 354: “higher-intensity form of chewing…” I think you mean vigilance instead of chewing.

Line 374: Distance to cover is only found to be significant in the SEQ region.

Line 395: remove the word social from (antipredator or social) as you did not find effects of factors on social vigilance.

Author Response

Point 1: Line 109: how specifically did group size influence the amount of time spent vigilant? Positive or negative?

Response 1: This has now been clarified, increasing group size decreased time spent vigilant

Point 2: Line 111: Same as above, what specific effect did food resources have?

Response 2: This has now been clarified

Point 3: Lines 111-113: This sentence is awkwardly placed before the main predictions. Consider rephrasing.

Response 3: This sentence has been removed and this paragraph rearranged

Point 4: Lines 113-116: Based on these predictions urban SEQ should exhibit the highest vigilance and non-urban ACT should exhibit the lowest. This could be expanded upon in the Discussion as the results do not support these predictions and this could be due to a variety of factors (as were mentioned by the authors). In sum, it’s interesting that the results deviate from the predictions.

Response 4: Text has been added and rearranged to discuss this further in the discussion section – Line 355

Point 5: Line 125: Add a comma between ABS and urban

Response 5: Added.

Point 6: Lines 136-137: what is meant by both observations? It says each site was visited only once.

Response 6: Observations has now been changed to seasons

Point 7: Line 157: Change “behavioural data was” to behavioural data were

Response 7: Changed

Point 8: Lines 180-184: The info in these sentences was already written above. Please delete.

Response 8: Deleted

Point 9: Line 188: add apostrophe to individuals head

Response 9: Added

Point 10: Line 199: I’m confused by the inclusion of scanning rate as this is not mentioned further in the analyses or results. Is this synonymous with vigilant events?

Response 10: Yes, scanning rate means number of vigilant events. Text edited for clarification

Point 11: Data Analysis section: Still a bit confusing. My interpretation is that the conditional inference tree was used to identify the variables to include in the linear mixed models. These variables were human population density, region, season, land use, rainfall, distance to cover, mob size and sex (as indicated on lines 211-212), and then model selection was done using AICc values. Is this correct? If not, then this section should be revised to explain.

Response 11: Yes, that is exactly what was carried out.

Point 12: Lines 215-216: Here the authors explain why they did not run a model on social vigilance (which makes sense), but do not explain why no models were used to evaluate vigilance while chewing verses not chewing. These behaviors were quantified and are displayed in the results section, but there is no explanation for why a statistical test was not performed to assess factors affecting these behaviors.

Response 12: This has now been clarified in the discussion (lines 301-303), as we were not aiming to differentiate between different types of vigilance.

Point 13: Lines 216-219: It’s not clear why this method was done as opposed to including a region*land use type interaction into the main model.

Response 13: This is explained the data analysis section Line 219

Point 14: Line 221: says a total of four models were run, but the supplementary show models 0-5.

Response 14: We ran 4 LMM, each of these had a different number of models in the model selection table, to obtain the best model.

Point 15: Lines 233-234: seems to be editing errors here: “(LMM: n=284 233 individuals (140 summer…”

Response 16: A bracket was missed, after winter. This has now been added

Point 17: Figure 2 y-axis: how is this in the hundreds, but in the thousands in fig 1 and fig 3?

Response 17: Fig 1 and 3 show the total time spent in each type of vigilance at each region but fig 2 shows the mean.

Point 18: Line 254: typo: (persons/km2) as at 2016).  Also parentheses inside of parentheses.

Response 18: The study period was 2016 so we have used HPD data from 2016

Point 19: Lines 304-306: Awkward sentence. Please re-phrase.

Response 19: This has been re-phrased

Point 20: Line 333: “However, a higher level of vigilance without chewing was found in the urban locations in the ACT…” This was not evaluated statistically and should be acknowledged.

Response 20: This has now been acknowledged and rephrased

Point 21: Line 354: “higher-intensity form of chewing…” I think you mean vigilance instead of chewing.

Response 21: Yes, this has been changed

Point 22: Line 374: Distance to cover is only found to be significant in the SEQ region.

Response 22: This has been added.

Point 23: Line 395: remove the word social from (antipredator or social) as you did not find effects of factors on social vigilance.

Response 23: Removed